# LANGUAGE CONTROLS MORE THAN TOP-DOWN ATTENTION: MODULATING BOTTOM-UP VISUAL PROCESSING WITH REFERRING EXPRESSIONS

## ABSTRACT

How to best integrate linguistic and perceptual processing in multimodal tasks is an important open problem. In this work we argue that the common technique of using language to direct visual attention over high-level visual features may not be optimal. Using language throughout the bottom-up visual pathway, going from pixels to high-level features, may be necessary. Our experiments on several English referring expression datasets show significant improvements when language is used to control the filters for bottom-up visual processing in addition to top-down attention.

## 1 INTRODUCTION

As human beings, we can easily understand the surrounding environment with our visual system and interact with each other using language. Since the work of Winograd (1972), developing a system that understands human language in a situated environment is one of the long-standing goals of artificial intelligence. Recent successes of deep learning studies in both language and vision domains have increased the interest in tasks that combine language and vision (Antol et al., 2015; Xu et al., 2015; Krishna et al., 2016; Suhr et al., 2017; Anderson et al., 2018b; Hudson & Manning, 2019). However, how to best integrate linguistic and perceptual processing is still an important open problem. In this work we investigate whether language should be used to control the filters for bottom-up visual processing as well as top-down attention.

In the human visual system, attention is driven by both *"top-down"* cognitive processes (*e.g.* focusing on target's color or location) and *"bottom-up"* salient, behaviourally relevant stimuli (*e.g.* fast moving objects) (Corbetta & Shulman, 2002; Connor et al., 2004; Theeuwes, 2010). Studies on embodied language explore the link between linguistic and perceptual representations (Pulvermüller, 1999; Vigliocco et al., 2004; Gallese & Lakoff, 2005) and it is often assumed that language has a *high-level* effect on perception and drives the *"top-down"* visual attention (Bloom, 2002; Jackendoff & Jackendoff, 2002; Dessalegn & Landau, 2008). However, recent studies from cognitive science point out that language comprehension also affects low-level visual processing (Meteyard et al., 2007; Boutonnet & Lupyan, 2015). Motivated by this, we propose a model[1] that can modulate either or both of *"bottom-up"* and *"top-down"* visual processing with language conditional filters.

Current deep learning systems for language-vision tasks typically start with low-level image processing that is not conditioned on language, then connect the language representation with high level visual features to control the visual focus. To integrate both modalities, concatenation (Malinowski et al., 2015), element-wise multiplication (Malinowski et al., 2015; Lu et al., 2016; Kim et al., 2016) or attention from language to vision (Xu et al., 2015; Xu & Saenko, 2016; Yang et al., 2016; Lu et al., 2017; Anderson et al., 2018a; Zellers et al., 2019) may be used. Specifically they do not condition low-level visual features on language. One exception is De Vries et al. (2017) which proposes conditioning the ResNet (He et al., 2016) image processing network with language conditioned batch normalization parameters at every stage. Our model differs from these architectures by having explicit *"bottom-up"* and *"top-down"* branches and allowing us to experiment with modulating one or both branches with language generated kernels.

---

[1]We will release our code and pre-trained models along with a reproducible environment after the blind review process.

We evaluate our proposed model on the task of *image segmentation from referring expressions* where given an image and a natural language description, the model returns a segmentation mask that marks the object(s) described. We can contrast this with purely image based object detection (Girshick, 2015; Ren et al., 2017) and semantic segmentation (Long et al., 2015; Ronneberger et al., 2015; Chen et al., 2017) tasks which are limited to predefined semantic classes. Our task gives users more flexibility to interact with the system by allowing them to describe objects of interest in free form language. The language input may contain various visual attributes (e.g., color, shape), spatial information (e.g., "on the right", "in front of"), actions (e.g., "running", "sitting") and interactions/relations between different objects (e.g., "arm of the chair that the cat is sitting in"). This makes the task both more challenging and suitable for comparing different strategies of language control.

The perceptual module of our model is based on the U-Net image segmentation architecture (Ronneberger et al., 2015). This architecture has clearly separated bottom-up and top-down branches which allows us to easily vary what parts are conditioned on language. The bottom-up branch starts from low level visual features and applies a sequence of contracting filters that result in successively higher level feature maps with lower spatial resolution. Following this is a top-down branch which takes the final low resolution feature map and applies a sequence of expanding filters that eventually result in a segmentation mask at the original image resolution. Information flows between branches through skip connections between contracting and expanding filters at the same level. We experiment with conditioning one or both of these branches with language.

To make visual processing conditional on language, we add language-conditional filters at each level of the architecture, similar to Misra et al. (2018). Our baseline only applies language-conditional filters on the top-down branch. Modulating only the top-down/expanding branch with language means the high level features extracted by the bottom-up/contracting branch cannot be language-conditional. Our model expands on this baseline by modulating both branches with language-conditional filters. Empirically, we find that adding language modulation to the bottom-up/contracting branch has a significant positive improvement on the baseline model. Our proposed model achieves state-of-the art performance on three different English referring expression datasets.

## 2 RELATED WORK

In this section, we review related work in several related areas: Semantic segmentation classifies the object category of each pixel in an image without language input. Referring expression comprehension locates a bounding box for the object(s) described in the language input. Image segmentation from referring expressions generates a segmentation mask for the object(s) described in the language input. We also cover work on language-conditional (dynamic) filters and studies that use them to modulate deep-learning models with language.

### 2.1 SEMANTIC SEGMENTATION

Primitive semantic segmentation models are based on Fully Convolutional Networks (FCN) (Long et al., 2015). DeepLab (Chen et al., 2017) and U-Net (Ronneberger et al., 2015) are the most notable state-of-the-art semantic segmentation models related to our work. DeepLab replaces regular convolutions with atrous (dilated) convolutions in the last residual block of ResNets (He et al., 2016) and implements Atrous Spatial Pyramid Pooling (ASPP) which fuses multi-scale visual information. The U-Net architecture (Ronneberger et al., 2015) improves over the standard FCN by connecting contracting (bottom-up) and expanding (top-down) paths at the same resolution: the output of the encoder layer at each level is passed to the decoder at the same level.

### 2.2 REFERRING EXPRESSION COMPREHENSION

Early models for this task were typically built using a hybrid LSTM-CNN architecture (Hu et al., 2016b; Mao et al., 2016). Newer models (Hu et al., 2017; Yu et al., 2016; 2018; Wang et al., 2019) use an Region-based CNN (R-CNN) variant (Girshick et al., 2014; Ren et al., 2017; He et al., 2017) as a sub-component to generate object proposals. Nagaraja et al. (2016) proposes a solution based on multiple instance learning. Cirik et al. (2018) implements a model based on Neural Module Networks (NMN) by using syntax information. Among the literature, Compositional Modular Network

(CMN) (Hu et al., 2017), Modular Attention Network (MAttNet) (Yu et al., 2018) and Neural Module Tree Networks (NMTree) (Liu et al., 2019) are the most notable state-of-the-art methods, and all of them are based on NMN (Andreas et al., 2016).

## 2.3 Image Segmentation from Referring Expressions

Notable models for this task include Recurrent Multimodal Interaction (RMI) model (Liu et al., 2017), Recurrent Refinement Networks (RRN) (Li et al., 2018), Dynamic Multimodal Network (DMN) (Margffoy-Tuay et al., 2018), Convolutional RNN with See-through-Text Embedding Pixel-wise heatmaps (Step-ConvRNN or ConvRNN-STEM) (Chen et al., 2019a), Caption-aware Consistent Segmentation Model (CAC) (Chen et al., 2019b), Bi-directional Relationship Inferring Network (BRINet) Hu et al. (2020) and Linguistic Structure guided Context Modelling (LSCM) module Hui et al. (2020). RRN which has a structure similar to U-Net, is built on top of a Convolutional LSTM (ConvLSTM) (SHI et al., 2015) network. Unlike our model, ConvLSTM filters are not generated from language representation and the multi-modal representation is used only in the initial time step. DMN generates 1 x 1 language-conditional filters for language representation of each word. It performs convolution operation on visual representation with language-conditional filters to generate multi-modal representation for each word. Like RMI, word-level multi-modal representations are fed as input to a multi-modal RRN to obtain multi-modal representation for image/language pairs. Step-ConvRNN starts with a visual-textual co-embedding and uses a ConvRNN to iteratively refine a heatmap for image segmentation. Step-ConvRNN uses a bottom-up and top-down approach similar to this work, however, our model uses spatial language generated kernels within a simpler architecture. CAC also generates 1 x 1 language-conditional dynamic filters. Unlike our model, CAC applies these dynamic filters to single resolution / single feature map and additionally generates location-specific dynamic filters (e.g. left, bottom) to capture relations between the objects exist at the different parts of the image. BRINet implements two different attention mechanisms: language-guided visual attention and vision-guided linguistic attention. LSCM implements a dependency parsing guided bottom-up attention mechanism to predict masks.

## 2.4 Language-conditional Filters

To control a deep learning model with language, early work such as Modulated ResNet (MOD-ERN) (De Vries et al., 2017) and Feature-wise Linear Modulation (FiLM) (Perez et al., 2018) used conditional batch normalization layers with only language-conditioned coefficients rather than customized filters. Finn et al. (2016) generates action-conditioned dynamic filters. Li et al. (2017) is the first work which generates dynamic language-conditional filters. Gao et al. (2018) proposes a VQA solution method which has a group convolutional layer whose filters are generated from the question input. Gavrilyuk et al. (2018) introduces a new task called as actor and action segmentation and to solve this task, proposes an architecture which uses dynamic filters for multiple resolutions. Similar to our work, Misra et al. (2018) adds language conditional filters to a U-Net based architecture for the task of mapping instructions to actions in virtual environments. **?** also uses an architecture based on U-Net and Misra et al. (2018) to solve a navigation and spatial reasoning problem. Those models only modulate top-down visual processing with language.

Referring expression models that incorporate language-conditional filters into the architecture include (Chen et al., 2019b; Margffoy-Tuay et al., 2018). Margffoy-Tuay et al. (2018) generates language-conditional filters for words individually rather than whole sentence. Chen et al. (2019b) generates 1 x 1 language-conditional filters from expressions. To make 1 x 1 language-conditional filters spatially aware, different filters are generated for different image regions (e.g. top, left, right, bottom).

Our main contribution in this work is an explicit evaluation of language conditional filters for bottom-up visual processing in comparison to only using language for top-down attention control.

## 3 Model

Figure 1 shows an overview of our proposed architecture. For a given referring expression $S$ and an input image $I$, the task is predicting a segmentation mask $M$ that covers the object(s) referred to. First, the model extracts a $64 \times 64 \times 1024$ tensor of low-level features using a backbone convolutional

neural network and encodes the referring expression $S$ to a vector representation $r$ using a long short-term memory (LSTM) network (Hochreiter & Schmidhuber, 1997). Starting with the visual feature tensor, the model generates feature maps in a contracting and an expanding path where the final map represents the segmentation mask, similar to U-Net (Ronneberger et al., 2015). 3x3 convolutional filters generated from the language representation $r$ (language kernels) are used to modulate both the contracting and the expanding paths. Our experiments show that modulating both paths improves the performance dramatically.

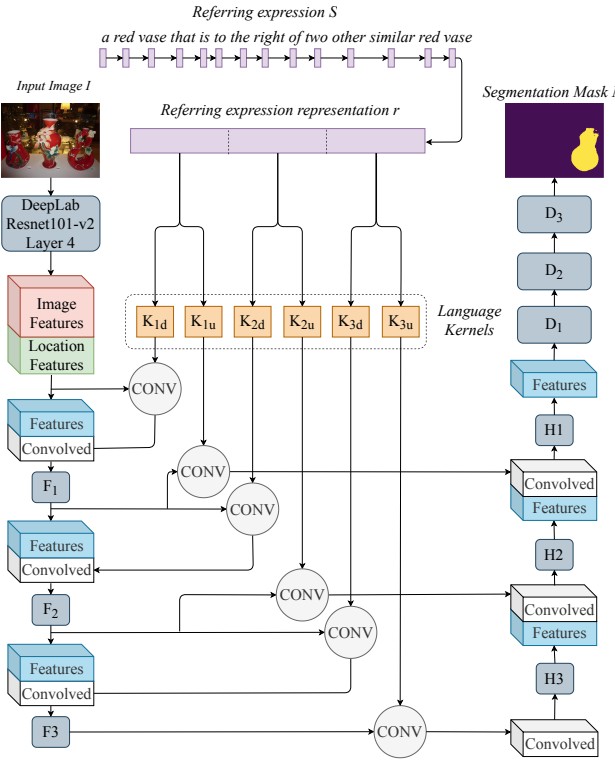

Figure 1: Overview of our model.

## 3.1 LOW-LEVEL IMAGE FEATURES

Given an input image $I$, we extract visual features from the fourth layer of the DeepLab ResNet101-v2 network (Chen et al., 2017) pre-trained on the Pascal VOC dataset (Everingham et al., 2010). We set $W = H = 512$ as the image size for our experiments. Thus, the output of the fourth convolutional layer of DeepLab ResNet101-v2 produces a feature map with the size of $(64, 64)$ and 1024 channels for this setup. We concatenate 8-D location features to this feature map following previous work (Hu et al., 2016b; Liu et al., 2017; Ye et al., 2019; Chen et al., 2019a). The final representation, $I_0$, has 1032 channels, and the spatial dimensions are $(64, 64)$.

## 3.2 LANGUAGE REPRESENTATION

Consider a referring expression $S = [w_1, w_2, ..., w_n]$ where $w_i$ represents the $i$'th word. In this work, each word $w_i$ is represented with a 300-dimensional GloVe embedding (Pennington et al., 2014), i.e. $w_i \in \mathbb{R}^{300}$. We map the referring expression $S$ to hidden states using a long short-term memory network (Hochreiter & Schmidhuber, 1997) as $h_i = LSTM(h_{i-1}, w_i)$. We use the final hidden state of the LSTM as the textual representation, $r = h_n$. We set the size of hidden states to 256, i.e. $h_i \in \mathbb{R}^{256}$.

### 3.3 SEGMENTATION MODEL

After generating image ($I_0$) and language ($r$) representations, our model generates a segmentation Mask $M$. We take the U-Net (Ronneberger et al., 2015) image segmentation model as the visual processing backbone. Our model extends the U-Net by conditioning both contracting and expanding branches on language using spatial language kernels.

Our model applies $m$ convolutional modules to the image representation $I_0$. Each module, $F_i$, takes the concatenation of the previously generated feature map ($Down_{i-1}$) and its convolved version with a $3 \times 3$ language kernel $K_{id}$ and produces an output feature map ($Down_i$). Each $F_i$ has a 2D convolution layer followed by batch normalization (Ioffe & Szegedy, 2015) and ReLU activation function (Maas et al., 2013). The convolution layers have $5 \times 5$ filters with $stride = 2$ and $padding = 2$ halving the spatial resolution, and they all have the same number of output channels.

Following Misra et al. (2018), we split the textual representation $r$ to $m$ equal parts ($t_i$) to generate language-conditional filters (language kernels). We use each $t_i$ to generate a language-conditional kernel ($K_{id}$):

$$K_{id} = \text{AFFINE}_i(\text{DROPOUT}(t_i)) \tag{1}$$

Each AFFINE$_i$ is an affine transformation followed by normalizing and reshaping to convert the output to a convolutional filter. DROPOUT is the dropout regularization (Srivastava et al., 2014). After obtaining the kernel, we convolve it over the feature map obtained from the previous module to relate expressions to image features:

$$G_{id} = \text{CONVOLVE}(K_{id}, Down_{i-1}) \tag{2}$$

Then, the concatenation of the resulting text-modulated features ($G_{id}$) and the previously generated features ($Down_{i-1}$) is fed into module $F_i$ for the next step.

In the expanding branch, we generate $m$ feature maps starting from the final output of the contracting branch as follows:

$$G_{ju} = \text{CONVOLVE}(K_{ju}, I_j) \tag{3}$$
$$Up_m = H_m(G_{mu}) \tag{4}$$
$$Up_j = H_j(G_{mu} \oplus Up_{j-1}) \tag{5}$$

Similar to the bottom-up phase, $G_{ju}$ is the modulated feature map with language-conditional kernels generated as follows:

$$K_{ju} = \text{AFFINE}_j(\text{DROPOUT}(t_j)) \tag{6}$$

where AFFINE$_j$ is again an affine transformation followed by normalizing and reshaping. Here, we convolve the kernel ($K_{ju}$) over the feature maps from the contracting branch ($Down_j$). Each upsampling module $H_m$ gets the concatenation ($\oplus$) of the text related features and the feature map ($Up_j$) generated from the previous module. Only the first module operates on just convolved features. Each $H_j$ consists of a 2D deconvolution layer followed by a batch normalization and ReLU activation function. The deconvolution layers have $5 \times 5$ filters with $stride = 2$ and $padding = 2$ doubling the spatial resolution, and they all have the same number of output channels.

After generating the final feature map $Up_1$, we apply a stack of layers ($D_1$, $D_2$, ..., $D_m$) to map $Up_1$ to the exact image size. Similar to upsampling modules, each $D_k$ is a 2D deconvolution layer followed by batch normalization and the ReLU activation. The deconvolutional layer has $5 \times 5$ filters with $stride = 2$ and $padding = 2$ to double the spatial sizes of the input. Each $D_k$ preserves the number of channels except for the last one which maps the features to a single channel for the mask prediction. There is no batch norm operation and the ReLU activation for the final module, instead we apply a sigmoid function to turn the final features into probabilities ($P \in \mathbb{R}^{H \times W}$).

### 3.4 LEARNING

Given the probabilities ($P \in \mathbb{R}^{H \times W}$) for each pixel belonging to the target object(s), and the ground-truth mask $G \in \mathbb{R}^{H \times W}$, the main training objective is the pixel-wise Binary-Cross-Entropy (BCE) loss:

Table 1: Ablation results and comparison with the previous works on the val set of UNC dataset with *prec@X* and *IoU* metrics.

| Method | prec@0.5 | prec@0.6 | prec@0.7 | prec@0.8 | prec@0.9 | IoU |
|---|---|---|---|---|---|---|
| Top-Down Modulation w/ FiLM layers | 60.97 | 53.19 | 43.71 | 31.62 | 10.57 | 54.21 |
| Top-Down Modulation w/ 1x1 filters | 63.00 | 54.20 | 44.71 | 32.01 | 10.74 | 55.04 |
| Top-Down Modulation w/ 3x3 filters | 64.29 | 55.26 | 46.29 | 32.96 | 11.78 | 56.13 |
| Bottom-Up Modulation (disconnected) w/ 3x3 filters | 69.92 | 62.84 | 52.16 | 32.70 | 7.18 | 58.77 |
| Bottom-Up Modulation w/ 3x3 filters | 72.13 | 65.92 | 57.93 | 43.80 | 17.49 | 60.73 |
| Dual Modulation w/ 1x1 filters | 71.76 | 65.77 | 58.19 | 44.80 | 17.05 | 60.75 |
| Dual Modulation w/ 3x3 filters (full model) | **73.53** | **67.53** | **60.00** | **46.96** | **18.80** | **61.95** |
| LSCM (Hui et al., 2020) | 70.84 | 63.82 | 53.67 | 38.69 | 12.06 | 61.54 |
| BRINet (Hu et al., 2020) | 71.83 | 65.05 | 55.64 | 39.36 | 11.21 | 61.35 |
| Step-ConvRNN (Chen et al., 2019a) | 70.15 | 63.37 | 53.15 | 36.53 | 10.45 | 59.13 |
| CMSA (Ye et al., 2019) | 66.44 | 59.70 | 50.77 | 35.52 | 10.96 | 58.32 |
| RRN (Li et al., 2018) | 61.66 | 52.5 | 42.4 | 28.13 | 8.51 | 55.33 |
| DMN (Margffoy-Tuay et al., 2018) | 65.83 | 57.82 | 46.80 | 27.64 | 5.12 | 54.83 |
| RMI (Liu et al., 2017) | 42.99 | 33.24 | 22.75 | 12.11 | 2.23 | 45.18 |

$$J = \frac{1}{HW} \sum_i^H \sum_j^W G_{ij} log(P_{ij}) + (1 - G_{ij}) log(1 - P_{ij}) \qquad (7)$$

## 4 EXPERIMENTS

In this section we first give the details of the datasets and our experimental configurations (Section 4.1). A detailed analysis of the contribution of our idea and the different parts of the architecture is given in Section 4.2. Then we present our main results and compare our model with the state-of-the-art (Section 4.3). Finally, Section 4.4 shows some qualitative results.

### 4.1 DATASETS AND EXPERIMENT SETUP

**Datasets:** We evaluate our model on and ReferIt (130.5k expressions, 19.9k images), UNC (142k expressions, 20k images), UNC+ (141.5k expressions, 20k images) (Yu et al., 2016) and Google-Ref (G-Ref) (104.5k expressions, 26.7k images) (Mao et al., 2016) (Kazemzadeh et al., 2014) datasets. Unlike UNC, location-specific expressions are excluded in UNC+ through enforcing annotators to describe objects by their appearance. ReferIt, UNC, UNC+ datasets are collected through a two-player game (Kazemzadeh et al., 2014) and have short expressions (avg. 4 words). G-Ref have longer and richer expressions, since its expressions are collected from Amazon Mechanical Turk instead of a two-player game. ReferIt images are collected from IAPR Tc-12 dataset (Escalante et al., 2010) and the others use images present in MS COCO dataset (Lin et al., 2014).

**Evaluation Metrics:** Following the previous work (Liu et al., 2017; Margffoy-Tuay et al., 2018; Ye et al., 2019; Chen et al., 2019a), we use overall intersection-over-union (*IoU*) and $precision@X$ as evaluation metrics. Given the predicted segmentation mask and the ground truth, the *IoU* metric is the ratio between the intersection and the union of the two. The overall *IoU* calculates the total intersection over total union score. The second metric, $precision@X$, calculates the percentage of test examples that have *IoU* score higher than the threshold $X$. In experiments, $X \in \{0.5, 0.6, 0.7, 0.8, 0.9\}$.

**Implementation Details:** As (Liu et al., 2017; Margffoy-Tuay et al., 2018; Ye et al., 2019; Chen et al., 2019a), we limit the maximum length of expressions to 20. In all convolutional layers, we set the filter size, stride, and number of filters ($ch$) as $(5, 5)$, 2, and 96, respectively. The depth is 4 in the U-Net part of the network. We set the dropout probability to 0.2 throughout the network. We use Adam optimizer (Kingma & Ba, 2014) with default parameters. We freeze the DeepLab ResNet101-v2 weights. There are 60 examples in each minibatch. We train our model for 15 epochs on a Tesla V100 GPU and each epoch takes at most two hours depending on the dataset.

Table 2: Comparison with the previous works on four datasets. Evaluation metric is the overall *IoU* and higher is better. Bold scores indicate the state-of-the-art performances. "-" indicates that the model has not been evaluated on the dataset. "n/a" indicates that splits are not same.

| Method | UNC | | | UNC+ | | | G-Ref | ReferIt |
|---|---|---|---|---|---|---|---|---|
| | val | testA | testB | val | testA | testB | val | test |
| CNN+LSTM (Hu et al., 2016a) | - | - | - | - | - | - | 28.14 | 48.03 |
| RMI (Liu et al., 2017) | 45.18 | 45.69 | 45.57 | 29.86 | 30.48 | 29.5 | 34.52 | 58.73 |
| DMN (Margffoy-Tuay et al., 2018) | 49.78 | 54.83 | 45.13 | 38.88 | 44.22 | 32.29 | 36.76 | 52.81 |
| DynamicFilters (Li et al., 2017) | - | - | - | - | - | - | - | 54.30 |
| KWA (Shi et al., 2018) | - | - | - | - | - | - | 36.92 | 59.09 |
| RRN (Li et al., 2018) | 55.33 | 57.26 | 53.93 | 39.75 | 42.15 | 36.11 | 36.45 | 63.63 |
| CMSA (Ye et al., 2019) | 58.32 | 60.61 | 55.09 | 43.76 | 47.6 | 37.89 | 39.98 | 63.80 |
| CAC (Chen et al., 2019b) | 58.90 | 61.77 | 53.81 | - | - | - | 44.32 | - |
| Step-ConvRNN (Chen et al., 2019a) | 60.04 | 63.46 | 57.97 | 48.19 | 52.33 | 40.41 | 46.4 | 64.13 |
| BRINet (Hu et al., 2020) | 61.35 | 63.37 | **59.57** | 48.57 | 52.87 | 42.13 | 48.04 | 63.46 |
| LSCM (Hui et al., 2020) | 61.47 | **64.99** | 59.55 | 49.34 | 53.12 | **43.50** | 48.05 | **66.57** |
| MAttNet (Yu et al., 2018) | 56.51 | 62.37 | 51.70 | 46.67 | 52.39 | 40.08 | n/a | - |
| NMTree (Liu et al., 2019) | 56.59 | 63.02 | 52.06 | 47.40 | 53.01 | 41.56 | n/a | - |
| Our Model | **61.95** | 63.85 | 58.14 | **50.42** | **54.16** | 42.15 | **49.76** | 64.63 |

## 4.2 ABLATION RESULTS

We performed ablation studies to better understand the contributions of the different parts of our model. Table 1 shows the performances of the different architectures on the UNC validation split with *prec@X* and overall *IoU* metrics. Unless otherwise specified, $3 \times 3$ language-conditional filters are used in our models.

**Modulating both top-down and bottom-up visual processing:** We implemented three models, Top-down Modulation, Bottom-Up Modulation and Dual Modulation, to show the effect of modulating language in expanding and contracting visual branches. Since language information leaks through cross-connections between visual branches, we also experimented with a bottom-up modulation model which has no connection between visual branches. Bottom-up Modulation outperforms Top-down Modulation with $\approx$4.6 IoU improvement. Modulating language in both visual branches yields the best results by improving Bottom-up Modulation model with $\approx$1.2 IoU score.

**Language-conditional Spatial Filters:** When we compare the performances of Top-Down Modulation w/ $1 \times 1$ filters and Top-Down Modulation models, we see that the usage of language-conditional spatial filters brings additional improvement over the base model. Similarly, if we use $1 \times 1$ filters in our full model, the performance of the model decreases significantly. We performed the same experiment on G-Ref dataset and observed $\approx$1.3 IoU difference again.

**FiLM layers vs. Language-conditional Filters**: Another method for modulating language is using conditional batch normalization De Vries et al. (2017) or its successor, FiLM layers. Thus, we also replaced language-conditional filters with FiLM layers in Top-Down Modulation w/ 1x1 model and observed $\approx$0.8 IoU improvement. Morever, since we can take advantage of language-conditional spatial filters, Top-Down Modulation w/ 3x3 model baseline outperforms its FiLM variation with $\approx$1.9 IoU improvement.

## 4.3 QUANTITATIVE RESULTS

Table 2 shows the comparison of our model with the previous work. Our model outperforms all previous models on all datasets. When we compare our model with the previous state-of-the-art model, Step-ConvRNN, the most significant improvement is on the G-Ref dataset.

We also compare our model with MAttNet and NMTree which are referring expression comprehension models. Since they present segmentation results after they predict bounding boxes for objects, they are comparable with our work. Our model is significantly better than MAttNet, NMTree which depends on an explicit object proposal network that is trained on more COCO images. This result shows the ability of our model to detect object regions and relate them with expressions.

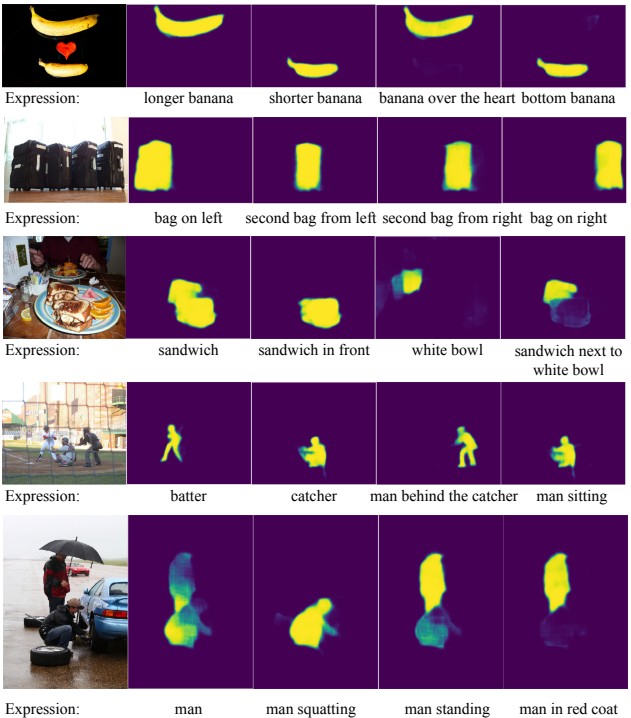

Figure 2: Some correct predictions of our model on UNC validation set. First column shows the input images and others show the predictions for the given referring expressions.

Table 1 presents the comparison of our model with the state-of-the-art in terms of $prec@X$ scores. The difference between our model and the state-of-the-art increases when the threshold increases. This indicates that our model is better at both finding and segmenting the referred objects.

## 4.4 QUALITATIVE RESULTS

In this section, we visualize some of the segmentation predictions of our model to gain better insights about the trained model.

Figure 2 shows some of the cases that our model segments correctly. These examples demonstrate that our model can learn a variety of language and visual reasoning patterns. For example, the first two examples of the first row show that our model learns to relate superlative adjectives (e.g., *longer*, *shorter*) with visual comparison. Examples include spatial prepositions (e.g., *on right, on left, next to, behind, over, bottom*) demonstrate the spatial reasoning ability of the model. We also see that the model can learn a domain-specific nomenclature (*catcher, batter*) that is present in the dataset. Lastly, we can see that the model can distinguish the different actions (e.g., *standing, squatting, sitting*).

Figure 3 shows some of the incorrect segmentation predictions from our model on the UNC validation dataset. In the figure, each group shows one of the observed patterns within the examples. One of them (a) is that our model tends to combine similar objects or their parts when they are hard to distinguish. Another reason for the errors is that some of the expressions are ambiguous (b), where there are multiple objects that could be the correspondence of the expression. And the model segments both possible objects. Some of the examples (d) are hard to segment completely due to the lack of light or objects that mask the referred objects. Finally, some of the annotations contain incorrect or incomplete ground-truth mask (c).

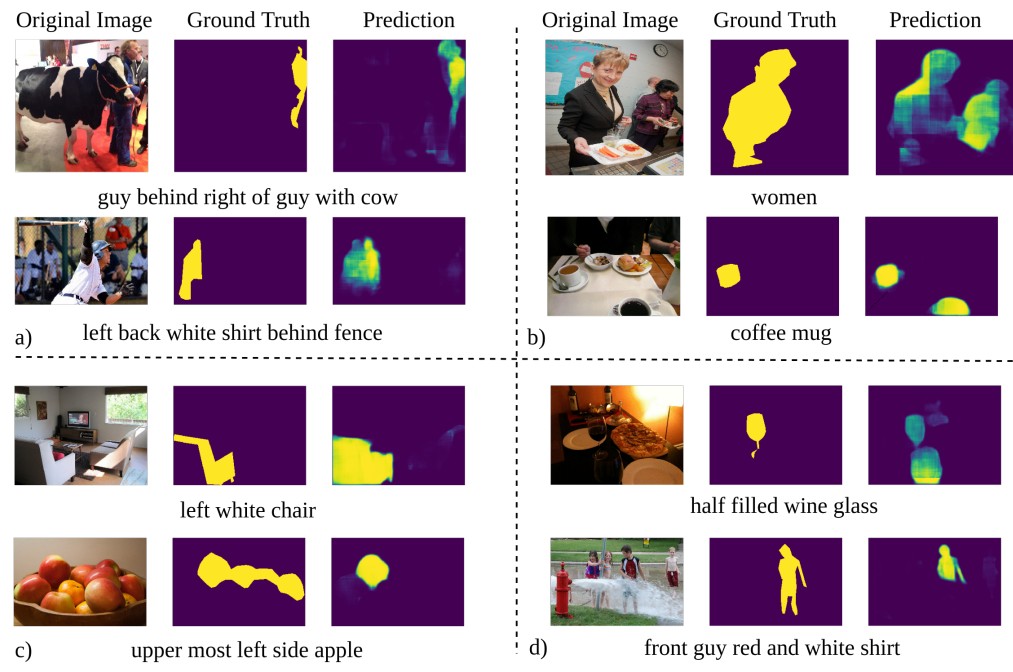

Figure 3: Some incorrect predictions of our model on UNC validation set. Each group (a-d) shows one pattern we observed within the predictions. In each group, the first column shows the original image, the second one is the ground truth mask and the third one is the prediction of our model.

## 5 CONCLUSION

We showed that modulating not only top-down but also bottom-up visual processing with language input improves the performance significantly. Our experiments showed that the proposed model achieves state-of-the-art results on 4 different benchmarks and performs significantly ($\approx 6$ IoU) better than a baseline which uses language only to direct top-down attention. Our future work will focus on using it as a sub-component to solve a far more language-vision task like mapping natural language instructions to sequences of actions.

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

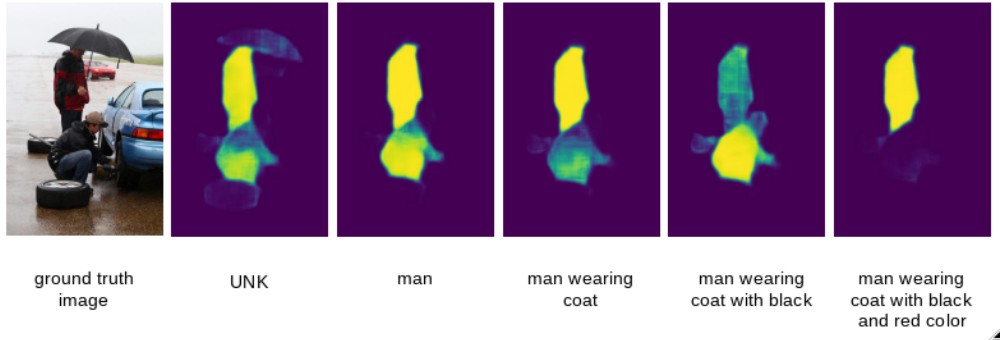

Figure 4: Incremental segmentation result of our model on UNC test split instance.

## A APPENDIX

### A.1 INCREMENTAL SEGMENTATION

We also analyzed the behaviour of our model with respect to incrementally given language input in Figure 4. In the initial step, our model only sees an unknown word token. In the second step, our model sees only the first word of the expression. In every step, our model starts to see a new word in addition to the previous ones. Figure 4 shows that our model can capture ambiguities in input image and expressions pairs. For unknown token input, our model captures all salient objects since there is no restriction. When the *man* word is fed, the model discards unrelated objects like umbrella and wheel. Additionally, when our model starts to see color words for coat, it initially focuses on both men, since both coats has black color. When it sees the final expression, it shifts its focus to the correct object.

