# OpenReview forum: "Language Controls More Than Top-Down Attention: Modulating Bottom-Up Visual Processing with Referring Expressions"
_ICLR.cc/2021/Conference — Reject_

### Official Review · AnonReviewer2 · 2020-10-28
**Possibly worthwhile idea, but poorly motivated and evaluated**

**Rating:** 2
**Confidence:** 4

**Review:**

This paper presents a model for image segmentation from referring expressions which integrates linguistic representations of the referring expressions both at low-level and high-level stages of visual processing. They argue that this model is both more cognitively plausible and more successful than models which only use linguistic representations to modulate attention over high-level visual features.

I vote for rejection, mainly on grounds of significance and quality, expanded below. To change my vote I would require a substantial improvement on one or both of the quality/significance issues listed below, either presenting the model with a clear conceptual motivation, or doing satisfactory model analysis to understand the contribution of the paper.

Pros: The presented model shows some moderate quantitative improvement over other recent work.

Cons: Poor conceptual motivation and error analysis, with little evident understanding of the actual effect of the linguistic representations within the model.

Quality/Significance

1. The paper does not provide a clear motivation for their model. Here are some arguments I looked for but did not find:
  1. Cognitively: There are some references to relevant cognitive science papers, but there seems to be little concrete inspiration taken from this or other cognitive work in the particular model design.
  2. A priori based on the task: What would we expect to gain from using language in early-stage visual representations? What sort of correlations might exist between particular types of linguistic input and low-level visual representations? This might be another way to motivate the model, but I can't find any such discussion in the introduction or anywhere else.
2. The evaluations don't convince me that this paper has made a significant conceptual contribution.
  1. The quantitative results don't seem to constitute an enormous improvement over past work. The variability in table 2 across evaluation sets makes me doubt the statistical significance of the claims. No statistical significance tests (across resamples of test data or across random training restarts) are provided.
  2. There is no satisfactory analysis of the actual cause of the model's success. What are the contents of the linguistic representations, and how exactly do they modulate low-level visual features? For reference, Hu et al. (2020, Figure 4) [1] and Hui et al. (2020, Figure 5) [2] both do some of what I'm looking for here, showing the influence of language on the behavior of the model. While the more complex representations used in this model make it more difficult to provide e.g. an easy heatmap, we absolutely need to see an error analysis that helps us believe your claim that language ought to play a role in low-level visual processing.

Originality

I don't closely follow the relevant literature and can't speak confidently on the originality of the model. I did have trouble understanding the innovation over Step-ConvRNN, however -- these models seemed within tweaking-distance of one another based on the presentation in this paper.

[1]: https://openaccess.thecvf.com/content_CVPR_2020/papers/Hu_Bi-Directional_Relationship_Inferring_Network_for_Referring_Image_Segmentation_CVPR_2020_paper.pdf#page=5
[2]: https://arxiv.org/pdf/2010.00515.pdf#page=14

## Post-rebuttal update

I have read the other reviews and the authors' rebuttals, and do not wish to change my review.

I strongly believe that numerical task improvements are not in themselves a conceptual contribution. I look forward to the results of the analyses the authors mention in response to R3-Q2, to better understand what exact interaction between language and low-level visual input is being modeled.

Along with R4 I remain unconvinced of the strength of the empirical results. The authors' response is not helpful here. I can't understand where the numbers (mean 60.74 IoU, std 0.06) come from -- taking stats across table 2 and table 1, I get very different results, so I must be misunderstanding where they come from.

Significance tests would not take too much time -- it's not absolutely critical that you retrain the models for this. You can use data resampling methods instead. For example, on each individual dataset, run bootstrap tests comparing the predictions of your model and others on random resamples of the evaluation data and corresponding predictions.
Pooling IoU results across datasets within model and then comparing between models can yield misleading results and should be avoided.

---

> ### Author Response · Authors · 2020-11-25
> **Thank you for your constructive comments**
>
> See the end of this section for an answer to 1-3.
>
> > The evaluations don't convince me that this paper has made a significant conceptual contribution.
>
> Our ablation analysis shows that modulating both bottom-up and top-down visual processing with language improves the performance over bottom-up-only (-1.22) IoU and top-down-only (-5.82 IoU) baselines. This is our main contribution in this study. We also showed that our model generalizes well enough on four different datasets achieving SOTA or near SOTA results.
>
> > The quantitative results don't seem to constitute an enormous improvement over past work. The variability in table 2 across evaluation sets makes me doubt the statistical significance of the claims. No statistical significance tests (across resamples of test data or across random training restarts) are provided.
>
> See the previous comment also. We agree with you but we didn't perform any signifiance tests because of two reasons: (i) it'd take too much time, (ii) no previous work presented these results. We spent our time performing comprehensive ablation studies to show our main contribution. We also performed several experiments for “Dual Modulation w/ 1x1 filters” model in ablation studies. Our model achieves 60.74 mean IoU for all the experiments with 0.06 standard deviation.
>
> > I don't closely follow the relevant literature and can't speak confidently on the originality of the model. I did have trouble understanding the innovation over Step-ConvRNN, however -- these models seemed within tweaking-distance of one another based on the presentation in this paper.
>
> The cognitive science studies we cited show that language has an important role in visual processing and has an effect on the early stages of the visual processing. Those studies measure this effect by looking at the response times to the inputs that trigger the visual cortex that are known to correspond to the early visual process. However, they do not discuss how language affects the visual perception in detail (nobody yet knows). We also do not possess the answer to this question. We were inspired by the idea of language having an effect on the low-level visual processing and tried this idea on a related task with a model that we can modulate both top-down and bottom-up visual processing with language explicitly.
> On the other hand, the Step-ConvRNN work does not show the effect of top-down and bottom-up visual processing individually. They only performed experiments with both  top-down and bottom-up visual processing at the same time. Their architecture is far more different/complicated than ours and it is also not suitable for this kind of ablation study.

---

### Official Review · AnonReviewer1 · 2020-10-29
**Very interesting paper**

**Rating:** 10
**Confidence:** 4

**Review:**

This article proposes a novel approach integrating language throughout the visual pathway for segmenting objects according to referring expressions.

The article is well written, and poses an important question about how best to integrate linguistic and visual information. The limitations of the currently dominant top-down approach are well argued. The answer proposed by the authors is to integrate linguistic information throughout the visual hierarchy. The task of segmenting by referring expression is important and well chosen.

The proposed model is sound, and well described in the article, and the experimental results demonstrate that the model outperforms clearly the state-of-the art in all metrics. The qualitative examples provided are quite impressive and demonstrate the success of the approach.

In sum, I feel this is a well written paper addressing a very timely and important problem in computer vision and AI research and should be of broad interest in the community.

---

> ### Author Response · Authors · 2020-11-25
> **Thank You**
>
> Thank you for your time and review. We believe that our work demonstrates the importance of modulating both bottom-up and top-down processing with language for vision-language studies.

---

### Official Review · AnonReviewer4 · 2020-10-30
**Interesting idea; empirical results are ok**

**Rating:** 4
**Confidence:** 3

**Review:**

This paper concerns the problem of image segmentation from referring expressions. Given an image and a query phrase about a particular object in the image, the goal is to locate the target object as a mask at the pixel level. The basic framework is U-Net, which consists of two branches: an image encoder and segmentation map decoder (connected at the bottom in a U-shape). The paper proposes to use language to modulate the image encoding and decoding process intensively, by applying auxiliary convolutional connections between the two branches and further condition the convolution kernel on the language embedding. Overall, the paper is easy to follow and has done a good literature review.

The major concern of the paper lies in the empirical results. Despite that the introduction of top-down and bottom-up language modulation significantly boost the baseline performance (Tab. 1), the full model struggles to match existing works on certain metrics such as UNC testA/testB, UNC+ testB, and ReferIt, which put a question mark on the effectiveness of the work. The results on the validation set are promising but not as good on the test set, which indicates a possible over-tuning of the model.

A minor comment on the model part. In the text above Eq. 1, the paper mentions "[...] ,we split the textual representation [...]". However, what is the rationale for splitting the representation since each split does not attach to any particular abstract of the image feature (low-level, mid-level, and high-level)?

Besides, some numbers from Tab. 1 do not match those from Tab. 2. For instance, the IoU on LSCM and Step-ConvRNN. Please double check.

============== Post-Rebuttal ==============

The authors' responses to point 1 & 2 do not sound (reflecting a question to another paper does not solve the problem). The authors mentioned "We made this decision based on Mei et al (2018) which proposed our baseline model (the top down approach)", where the reference of Mei et al (2018) cannot be found in the paper, as a critical baseline. This raises a flag on the novelty of the work and completeness of the related work. Therefore, I am lowering my rating to 4.

---

> ### Author Response · Authors · 2020-11-25
> **Thank you for your constructive comments**
>
> > The major concern of the paper lies in the empirical results. Despite that the introduction of top-down and bottom-up language modulation significantly boost the baseline performance (Tab. 1), the full model struggles to match existing works on certain metrics such as UNC testA/testB, UNC+ testB, and ReferIt, which put a question mark on the effectiveness of the work. The results on the validation set are promising but not as good on the test set, which indicates a possible over-tuning of the model.
>
> The main point of our paper is to argue for the effectiveness of modulating both top-down and bottom-up visual streams with language, which our results support. We do not believe there was any over-tuning: we only tuned the model (e.g., the number of layers, the number of filters, the size of the LSTM) looking at the results obtained on the UNC validation set. We use the other validation sets only for early stopping. This will be made more clear in the camera-ready version.
>
> > A minor comment on the model part. In the text above Eq. 1, the paper mentions "[...] ,we split the textual representation [...]". However, what is the rationale for splitting the representation since each split does not attach to any particular abstract of the image feature (low-level, mid-level, and high-level)?
>
> We made this decision based on Mei et al (2018) which proposed our baseline model (the top down approach). We tried to use the final hidden state as a whole for language kernel generation in our preliminary experiments, however, we observed slight declines in the performance.
>
> > Besides, some numbers from Tab. 1 do not match those from Tab. 2. For instance, the IoU on LSCM and Step-ConvRNN. Please double check.
>
> We have obtained those numbers from the corresponding studies. Step-ConvRNN presents results of different models (step=4 and step=5) for the ablation study and the SOTA comparison. We checked it again and we couldn’t find a reason for why the authors present different numbers for LSCM.

---

### Official Review · AnonReviewer3 · 2020-11-03
**good experimental setup, lacks the depth of novelty and analyses**

**Rating:** 5
**Confidence:** 4

**Review:**

This paper proposes to integrate visual and linguistic features in both top-down and bottom-up modulation of the visual input. This is done by fusing two modalities while doing convolution and deconvolution operations over the visual input. Experiments on image segmentation from referring expressions in standard datasets show that the proposed approach achieves state-of-the-art or competitive results. Ablation studies show that both top-down and bottom-up is essential. I believe the novelty and contribution are rather thin because many ways of the modeling language are not explored at all.

Below I list suggestions (S) and questions (q) for authors:

S1 Second paragraph of introductions: please add a figure to explain the concepts of top-down, bottom-up processing, high-level, low-level effects etc.

S2 Section 2.2.: Please cite the below papers [1,2] for referring expression comprehension. For Section 2.4 please add [3]

S3 Figure1: following this figure is not intuitive. I recommend adding two arrows for top-down and bottom-up processing and adding more space between two branches.

S4 Section 4.2: It is not clear how each of these ablations was performed. For instance, I'm not 100% sure whether two modalities are fused at different levels of top-down or bottom-up processing.

[1] Nagaraja et. al. "Modeling context between objects for referring expression understanding."
[2] Cirik et. al "Using syntax to ground referring expressions in natural images."
[3] Chen et. al. "Touchdown: Natural language navigation and spatial reasoning in visual street environments."

S4: Section 4.3: the claim of modeling the long-range dependencies is a bit speculative. I would rephrase that.

S5: Figure2: Failure cases are more informative than successful ones. Please either bring the figure from A.1 or add a comparison with a model from the literature where the other model is successful where yours is not to do a contrastive analysis on how your model can be improved.

Q1: Section3: What's the effect of the number of layers for the model? Why stop at 3? Do you have results for the number of layers 0,1,2?

Q2: Is there a way to interpret the interaction between language and visual input?

Q3: Have you experimented with different ways of fusing or processing language input? Examples: gating the language representation, attention over tokens, using different fusion methods, bi-directional LSTM, BERT-like contextual representations, adding inductive bias with parse tree for referring expressions, alignment between feature maps and word tokens or phrases?

---

> ### Author Response · Authors · 2020-11-25
> **Thank you for your suggestions**
>
> Thank you for your suggestions. We implemented S2, S3 and S5 for right now.
>
> Answer to Q1: We obtained results for the number of layers 2, 3 and 4. The depth=4 improves the performance slightly (1 IoU) over depth=3. In this architecture, the contradicting branch halves the input on each layer, which limits the number of layers that can be used in the model. Due to the size limit of the GPU, we haven’t experimented with larger number of layers.
>
> Answer to Q2: One of the possible ways of interpreting the interaction between language and visual processing is clustering a specific layer’s language filters  obtained for each phrase. Possibly, obtained clusters would give information about which language component (adjectives, prepositions, nouns etc.) has a role on which part of the architecture. Another possible way of inspecting the effect of language on the visual processing could be a word removing/masking experiment. In this experiment, checking the segmentation performance of the model after removing/masking a word would give some insights whether the model actually uses the words given in the phrase or not. Currently, we are working on both analyses. We also added an incremental analysis in the appendix.
>
> Answer to Q3: We have experimented with bi-directional LSTM and self-attention over embeddings obtained from the last layer of a Bert model. In our preliminary experiments, these approaches didn’t improve the current model. Due to the memory constraints, we continued with the basic LSTM model. As suggested by the question, introducing an inductive bias by aligning feature maps and word tokens using the parse tree of the expression could improve the performance of the model or help the learning process. However, it also requires an understanding of how language works in visual processing (which part of the expression affects which part of the visual processing). In this study, we proposed an end-to-end approach where the model itself learns the connection between language and visual processing.

---

### Decision · Program_Chairs · 2021-01-07
**Final Decision**

**Decision:**

Reject

**Comment:**

The paper proposes to improve image segmentation from referring expression  by integrating visual and language features using an UNet architecture and experimenting with top-down, bottom-up, and combined (dual) modulation.

Review Summary: The submission received divergent reviews with scores spanning from 2 (R2) to 5 (R3,R4) to 10 (R1).  The author response failed to address the reviewer concerns with some reviewer (R4) lowering their score tto 4 after the rebuttal.  It also became clear that some relevant work (Mei et al, 2018) was used for the baseline but not cited.  The author response also did not recognize the importance of significance tests.

As there is considerable work in the area of image segmentation from referring expression, and the proposed model is very similar to the LingUNet model of Misra 2018, the originality and significance of the work is fairly low.  The main contributions appears to be experimental comparisons of the three types of modulation (top-down, bottom-up, dual).

Pros:
- Investigation of a important problem of grounding language to visual regions
- Experimental study of whether dual modulation improves image segmentation from referring expression

Cons
- Relatively minor novelty with limited analyses (R3,R2)
- Missing citations (see R3's comments).  Relevant work (Mei et al, 2018) which was the basis for the top-down baseline model, was used but not cited or properly compared against
- Relatively weak experimental results (R2,R4).  As R4 noted, while validation results are good, test results are weak compared to existing work, indicating potential overtuning.
- No qualitative comparisons against baselines.
- Cognitive claims not backed up and limited discussion/analysis (R2)

Recommendation:
The AC concurs with R2, R3, and R4 that the work is limited in novelty and not ready for publication at ICLR.   Despite R1's high score, referring expression for image segmentation is a well studied task, and it is unclear what are the key innovations of the proposed model over LingUNet.  Due to the limited novelty, relatively weak test results, as well as other flaws pointed out by the reviewers, the AC recommends rejection.